# Simulation of Vancomycin Exposure Using Trough and Peak Levels Achieves the Target Area under the Steady-State Concentration–Time Curve in ICU Patients

**DOI:** 10.3390/antibiotics12071113

**Published:** 2023-06-27

**Authors:** Yuta Ibe, Tomoyuki Ishigo, Satoshi Fujii, Satoshi Takahashi, Masahide Fukudo, Hideki Sato

**Affiliations:** 1Department of Pharmacy, Sapporo Medical University Hospital, Sapporo 060-8543, Japan; 2Department of Infection Control and Laboratory Medicine, Sapporo Medical University School of Medicine, Sapporo 060-8543, Japan; 3Department of Pharmacotherapeutics, Faculty of Pharmaceutical Sciences, Hokkaido University of Science, Sapporo 006-8585, Japan

**Keywords:** area under the concentration–time curve, critically ill patients, intensive care unit, therapeutic drug monitoring, vancomycin

## Abstract

The therapeutic drug monitoring (TDM) of vancomycin (VCM) in critically ill patients often results in the estimated area being under the concentration–time curve (AUC) values that deviate from individual observations. In this study, we investigated the factors influencing the achievement of the target AUC of VCM at steady-state in critically ill patients. We retrospectively collected data from patients treated with VCM in an intensive care unit (ICU). Multivariate analysis was used to adjust for significant factors with *p* < 0.05 and identify new factors affecting the achievement of the target AUC at steady-state for VCM. Among the 113 patients included in this study, 72 (64%) were in the 1-point group (trough only), whereas 41 (36%) were in the 2-point group (trough/peak). The percentage of patients achieving the target AUC at the follow-up TDM evaluation was significantly higher in the two-point group. Multivariate analysis showed that being in the 2-point group and those with a 20% or more increase (or decrease) in creatinine clearance (CCr) were both significantly associated with the success rate of achieving the target AUC at the follow-up TDM. Novel findings revealed that in patients admitted to the ICU, changes in renal function were a predictor of AUC deviation, with a 20% or more increase (or decrease) in CCr being an indicator. We believe the indicators obtained in this study are simple and can be applied clinically in many facilities. If changes in renal function are anticipated, we recommend an AUC evaluation of VCM with a two-point blood collection, close monitoring of renal function, and dose adjustment based on reanalyzing the serum concentrations of VCM.

## 1. Introduction

The practice guidelines for the therapeutic drug monitoring (TDM) of vancomycin (VCM) recommend an area under the concentration time curve (AUC) /minimum inhibitory concentration (MIC) ratio of 400–600 μg·h/mL to predict clinical efficacy and adverse effects [1,2]. AUC-guided TDM monitoring is essential in critically ill patients requiring acute-phase and intensive care, where the pharmacokinetics (PK) of VCM seem unstable. Most patients admitted to the intensive care unit (ICU) have unstable hemodynamic and renal function, including those exhibiting severe infections [3,4,5], renal dysfunction [6], and burns [7,8]. Therefore, there may be a discrepancy between the serum concentration of VCM estimated via the simulation and the actual measured serum concentration.

Severely ill patients admitted to the ICU are at a high risk of developing acute kidney injury (AKI), especially patients with sepsis and septic shock [9]. Furthermore, a recent report recommends an AUC-guided dosing design for VCM, since ICU admissions are an independent risk factor for AKI in patients receiving VCM [10,11].

Although it has been considered preferable to obtain two-concentration samples (peak/trough) to accurately estimate the AUC using the Bayesian approach, updated TDM guidelines [1,2] suggest that using only the trough may be sufficient to estimate the AUC in patients with mild to moderate disease. Since predicting the true AUC from trough-only data in critically ill patients is difficult, due to changes in pathology and renal function, two-point measurements may be required for an improved assessment of these cases [12].

To evaluate the ability of AUC estimation, a recent study examined the performance of AUC estimation with a population PK model based on the two-point sample compared with the reference AUC calculated according to the log-linear trapezoidal rule using eight measured drug concentrations after a single intravenous infusion [13]. VCM was administered at 12 h intervals, and the AUC achieved by 1-point testing was approximately 70%, whereas the AUC achieved by 2-point testing was approximately 82% [13]. Furthermore, the estimates’ accuracy decreased when VCM was administered every 24 h. Considering the decrease in accuracy, we hypothesize that dosing designs using estimated AUCs from trough-only samples may deviate from the actual steady-state AUC. Furthermore, the identification of novel factors associated with achieving the target AUC is important for successful VCM treatment.

We believe that more data with AUC-guided monitoring of VCM in critically ill patients are needed to avoid adverse effects and improve therapeutic efficacy using VCM. In this study, we investigated the AUC estimation during the initial TDM, compared one-point or two-point blood collection groups, and evaluated the success rate of achieving the target AUC value at the steady-state. Furthermore, we studied factors associated with the achievement of the target AUC values.

## 2. Methods

### 2.1. Study Design and Participants

This retrospective study included patients admitted to the ICU, who were treated with VCM at the Sapporo Medical University Hospital between January 2017 and March 2023. The inclusion criteria were as follows: (1) age ≥ 18 years, (2) receipt of at least 5 doses of VCM, and (3) completion of initial and follow-up TDM evaluations. The exclusion criteria were as follows: (1) age <18 years, (2) completion of VCM treatment prior to the follow-up TDM evaluation, (3) TDM prior to ICU admission, (4) measurement of VCM troughs at non-steady-state timing, (5) prophylactic administration after surgery, (6) ICU discharge within 72 h, (7) receipt of hemodialysis, and (8) missing data.

### 2.2. VCM Dosing and Pharmacodynamics Data

The TDM of the VCM in this study was performed in all cases by pharmacists in the ICU wards, and the AUC was estimated using software. Furthermore, based on the TDM analysis, the dosage of VCM, dosing interval, and timing of blood sample collection was proposed.

The initial dose of VCM was determined, based on either physician-initiated dose determination or a pharmacist’s suggestion after the initial dose design. The maintenance dose after the initial TDM was determined by a pharmacist who conducted the TDM assessment and adjusted the dose using simulation software. Blood samples were collected immediately before VCM administration to obtain trough concentrations. Samples for peak concentrations were collected 1–2 h after an intravenous VCM infusion. Serum VCM was measured, using Abbott Japan’s Architect vancomycin ST^®^. The lower limit of quantitation for the analyte was 0.3 μg/mL, and intra-assay reproducibility was less than 5%. The AUC values were calculated from the trough only (1-point group) or trough and peak (2-point group). The timing of the initial TDM was determined according to the trough values from the second to seventh VCM administrations for the 1-point group and trough and peak values around the second to fifth administrations for the 2-point group. Follow-up TDM was evaluated in patients who received at least five doses of VCM. In all cases, pharmacists performed TDM of vancomycin in the ICU ward at our hospital and suggested changes in the dosage, dosing interval, and timing of blood collection. All PK calculations and modeling were performed using practical AUC-guided TDM for VCM (PAT) [13].

Laboratory findings, vital sign data, sequential organ failure assessment (SOFA) scores [14], and acute physiology and chronic health inquiry (APACHE) II scores [15] were collected from medical records at the initial VCM dose. Renal function was measured continuously from the initiation of VCM and estimated using the most recent serum creatinine level. Creatinine clearance (CCr) was calculated using the Cockcroft–Gault formula (Equation (1)) [16]. The eGFRcre values were calculated, based on equations provided by the Japanese Society of Nephrology (Equation (2)) [17]. The definitions of sepsis and septic shock followed “The Third International Consensus Definitions for Sepsis and Septic Shock” (Sepsis-3) [18].
(1)$$\mathrm{CCr}\;\left(\mathrm{mL}/\min\right)=\frac{\left(140-\mathrm{Age}\right)\times \mathrm{Body}\;\mathrm{Weight}\;\left(\mathrm{kg}\right)}{72\times \mathrm{Scr}\;\left(\mathrm{mg}/\mathrm{dL}\right)}\;\left(\times 0.85\;\mathrm{if}\;\mathrm{female}\right)$$
(2)$$\;\mathrm{eGFRcre}\;\left(\mathrm{mL}/\min/1.73{\;m}^{2}\right)=194\times {\mathrm{Scr}}^{-1.094}\times {\mathrm{Age}}^{-0.287}\;\left(\times 0.739\;\mathrm{if}\;\mathrm{female}\right)$$

### 2.3. Outcomes

The primary endpoint of this study was the success rate in achieving the target AUC values (400–600 μg·h/mL) at the follow-up TDM. The achievement rate of the target AUC values was compared between the 1-point and 2-point blood collection groups.

Secondary outcomes were investigated as factors related to the rate of the achievement of the target AUC values: changes in renal function, number of days to follow-up TDM, and the presence of concomitant medications. In addition, for changes in renal function, patient backgrounds were compared between groups with and without changes in renal function.

### 2.4. Statistical Analysis

Categorical variables were expressed as percentages, and the continuous numerical data were expressed as the mean ± standard deviation (SD), median (interquartile range [IQR] 25th–75th percentile), and frequencies (percentages). We used the *t*-test or the unpaired Mann–Whitney U test to compare parametric data and non-parametric data, respectively, between the 1-point and 2-point groups. For categorical data, groups were compared using the χ^2^ test or Fisher’s exact test, as appropriate. To predict the achievement rate of the target AUC at the follow-up TDM, the optimal cutoff value for changes in CCr between the initial and follow-up TDM and days from the initial TDM to follow-up TDM were calculated. The optimal cutoff value was determined, based on the Youden index. For univariate and multivariate analyses, a logistic regression analysis was used to calculate the adjusted odds ratio (aOR) with a 95% confidence interval (CIs). Explanatory variables were selected, based on their statistical significance (*p* < 0.05) in the univariate analysis. Statistical significance was set at *p* < 0.05. All analyses were performed using JMP^®^ 15 software (SAS Institute Inc., Cary, NC, USA).

## 3. Results

### 3.1. Study Participant Comparison

The enrollment protocol and exclusion criteria for this study are shown in Figure 1. Of the 298 patients who received VCM upon admission to the ICU, only 113 were included in this study whereas 185 were excluded based on the exclusion criteria. A total of 72 patients were included in the 1-point blood collection group (1-point group), and 41 patients were included in the 2-point blood collection group (2-point group) (Figure 1). The baseline patient characteristics are presented in Table 1. The mean age was 64.8 ± 13.0 years; 33 patients (29.2%) were female; the mean body mass index (BMI) was 23.3 ± 4.1 kg/m^2^; and the proportion of BMI ≥25 kg/m^2^ was 26.5%. The median SOFA score was 7 points (IQR 4–11 points), and the APACHE II score was 21 points (IQR 15–28 points). The median baselines of CCr and eGFRcre were 73.3 mL/min (IQR 44.8–104.9 mL/min) and 71.2 mL/min/1.73 m^2^ (IQR 43.9–98.3 mL/min/1.73 m^2^), respectively.

Among the cases analyzed, loop diuretics were administered in 58 cases (51%), spironolactone in 21 (19%), tolvaptan in 7 (6%), and acetazolamide in 1 (0.8%). Diuretics are frequently prescribed as a monotherapy, or in conjunction with loop diuretics. The catecholamines included noradrenaline in 50 cases (44%), dobutamine in 17 (15%), and dopamine in 5 (5%). Noradrenaline was predominantly employed as a standalone agent, or in combination with other catecholamines.

In a comparison between the 1-point and 2-point groups, the SOFA score (median (IQR); 7 (3–9) vs. 9 (6–11) points, *p* = 0.037), blood urea nitrogen (BUN) (21 (16–34) vs. 30 (24–55) points, *p* = 0.005), catecholamine use (40.3% vs. 65.9%, *p* = 0.009), diuretic use (45.8% vs. 65.9%, *p* = 0.040) were significantly higher in the 2-point group than in the 1-point group (Table 1). A comparison of VCM doses and days from TDM between the 2 groups showed significant differences in the loading dose (15.3% vs. 46.3%, *p* < 0.001), days to the initial TDM (2 (2,3) vs. 1 (1,2), *p* < 0.001), and days from the initial TDM to follow-up TDM (4 (4,5) vs. 4 (3,4), *p* < 0.001). No significant differences in age, sex, BMI, complications, baseline CCr, APACHE II scores, the maintenance dose of VCM, and the dosing interval for VCM were observed between the 1-point and 2-point groups. Since the study additionally included overlapping gram-positive and -negative bacterial infections, the patients received multiple antibiotics. Regardless, there was no significant difference in the percentage of each antibiotic used in the two groups.

### 3.2. Achievement Rate of Target Area under the Curve

In this study, the achievement rate of the target AUC at the follow-up TDM was 61.1% (69/113) for all patients. There was no significant difference in the achievement rate of the target AUC at initial TDM between the 2 groups (54% vs. 71%, *p* = 0.084) (Figure 2a). In contrast, the 2-point group achieved the target AUC at the follow-up TDM at a significantly higher rate than the 1-point group (51% vs. 78%, *p* = 0.005) (Figure 2b).

### 3.3. Relationship between Changes in Renal Function and Area under the Curve Deviation

The correlation between the rate of change in renal function from the initial and follow-up TDM and the rate of deviation of the AUC at the initial and follow-up TDM, are shown in Figure 3. There was a significant negative correlation between the rate of change in renal function and the rate of deviation of the AUC (weak correlation, R^2^ = 0.234, r = −0.345, *p* < 0.001).

Based on these results, the optimal cutoff values related to the achievement rate of the target AUC at the follow-up TDM were calculated (Appendix A). The cutoff value for changes in CCr between the initial and follow-up TDM was 18.1% (AUC, 0.63; sensitivity, 0.77%; and specificity, 0.50%). Similarly, the cutoff value for the number of days from the initial to follow-up TDM was 4 days (AUC, 0.66; sensitivity, 0.77%; and specificity, 0.48%) (Appendix A). Because it was not our purpose to obtain an accurate estimate of the cutoff value, the changes in CCr were approximated as ±20%. A change in CCr of more than ±20% indicates an increase (or decrease) of at least 20% in CCr between the initial TDM and follow-up TDM. The correlation between CCr at the initial and follow-up TDMs, are shown in Appendix A.

### 3.4. Logistic Regression Analysis of Factors Associated with Achievement Rate of Target Area under the Curve at the Follow-Up Therapeutic Drug Monitoring

In the univariate analysis, the 2-point group (OR: 3.36, 95% CI: 1.41–8.04, *p* = 0.006), a 20% or more increase (or decrease) in CCr (OR: 0.37, 95% CI: 0.16–0.84, *p* = 0.017), >4 days between the initial and follow-up TDM (OR: 0.33, 95% CI: 0.15–0.75, *p* = 0.008), and diuretics use (OR: 2.25, 95% CI: 1.04–4.86, *p* = 0.040) were all significantly associated with the achievement rate of the target AUC (Table 2). In the multivariate logistic regression analysis, based on the results of the univariate analysis adjusted to force the inclusion of factors with *p* < 0.05, the 2-point group (OR: 2.89, 95% CI: 1.06–7.84, *p* = 0.038) and a 20% or more increase (or decrease) in CCr (OR: 0.25, 95% CI: 0.09–0.67, *p* = 0.006) were independent predictors of achieving the target AUC (Table 2).

### 3.5. Characteristics of Patients with or without Changes in Creatinine Clearance

The patient characteristics are shown for the groups in which the changes in CCr between the initial TDM and follow-up TDM were a 20% or more increase (or decrease) (Table 3). There were 34 patients in the group with a 20% or more increase (or decrease) and 79 patients in the group without changes. There were no significant differences in age, BMI, SOFA score, or APACHE II score between the groups with or without CCr changes (Table 3). The percentage of patients with complications was additionally similar between the two groups. In contrast, patients in the group with a 20% or more increase (or decrease) were more likely to be female (22% vs. 47%), be diagnosed with sepsis (47% vs. 71%) or septic shock (35% vs. 56%), and be undergoing continuous renal replacement therapy (CRRT) use (19% vs. 38%) compared to patients without CCr changes. The CCr changes group showed decreased renal function; creatinine (IQR: 0.7 (0.6–1.0) vs. 1.0 (0.6–1.5) mg/dL) and BUN (IQR: 22 (16–34) vs. 34 (22–54) mg/dL) were higher; and CCr (IQR: 76.4 (59.9–108.3) vs. 49.6 (33.0–87.5) mL/min) and eGFRcre (IQR: 76.8 (52.2–101.4) vs. 53.3 (30.4–72.9) mL/min/1.73 m^2^) were significantly lower. The use of catecholamine (43% vs. 65%) and vasopressin (5% vs. 21%) were higher in the group with changes in CCr than in the group without changes in CCr. No significant differences were observed for VCM treatment or AUC values.

## 4. Discussion

In this study, we found that the estimation of AUC by a 2-point blood collection of trough and peak values at the initial TDM in critically ill patients enhances the achievement of target AUC values at steady-state. Furthermore, we identified changes in renal function, especially a 20% or more increase (or decrease) in CCr, as a novel risk factor associated with a failure to achieve the target AUC. If changes in renal function are anticipated in critically ill patients, we recommend an AUC evaluation with a 2-point blood collection, an enhanced monitoring of renal function, and a dose adjustment based on the retesting of blood VCM levels.

Because VCM is a renally excretory drug, more than 90% of the dose is excreted in the urine by glomerular filtration in patients with normal renal function [19]. The dose of VCM is adjusted according to the patient’s renal function, and the target serum concentration of VCM is AUC/MIC 400 to 600 μg·h/mL for efficacy and safety reasons [1,2]. The accuracy of this AUC estimation is higher for peak and trough measurements than trough-only PK sampling [12]. However, changes in renal function over time should be considered in order to achieve target AUC values at steady-state. For renally excreted drugs such as VCM, changes in renal function significantly impact blood levels; however, there is currently no clear clinically applicable indicator. Although there have been many studies on the relationship between the blood levels of VCM and renal function [6,10,20,21], there may be a discrepancy between the individual AUC values of VCM and those estimated by TDM simulation in critically ill patients, where changes in renal function and drug transfer in the body are difficult to predict. This discrepancy in AUC values may be due to changes in the clearance of VCM, resulting from changes in renal function over time. Therefore, we examined the relationship between the rate of changes in renal function and AUC values to avoid steady-state AUC value deviations associated with changes in renal function. In this present study, the rate of changes in renal function and the rate of AUC discrepancy at the initial and follow-up TDM were negatively correlated (Figure 3). Furthermore, a 20% or more increase (or decrease) in CCr from the baseline was associated with an inability to achieve the target AUC at the follow-up TDM (Appendix A). A 20% or more increase (or decrease) in CCr was an additional independent predictor of the non-achievement of the target AUC in a multivariate analysis. Based on these results, we identified a 20% or more increase (or decrease) in CCr, as a novel risk factor associated with the non-achievement of the target AUC. Previous reports have not examined measures to predict vancomycin AUC deviations, and our index provides a new benchmark for patients admitted to the ICU. It should be noted, however, that the indicator “20% or more increase (or decrease) in CCr” had low specificity. This means that changes in renal function do not necessarily predict AUC deviations in all ICU patients. However, the calculation of CCr is simple, and we believe that observing its variation is an indicator that can be easily applied clinically.

An additional analysis examined the clinical characteristics of patients with a 20% or more increase (or decrease) in CCr. In a previous study, about half of the patients admitted to the ICU developed AKI, with sepsis being the most common etiology at 40–50% [9,22,23]. In this study, 36/61 (59%) patients diagnosed with sepsis had AKI or were on continuous renal replacement therapy. This could additionally explain the lower renal function in the group with a 20% or more increase (or decrease) in CCr. On the other hand, sepsis-associated AKI has been shown to improve 48 h after onset [24]. Therefore, the pathogenesis of sepsis should be of particular concern in inferring changes in renal function during VCM administration.

Circulatory agonists are additionally used when sepsis presents with hypotension; however, the use of circulatory agonists additionally affects renal function [25,26]. Norepinephrine administration was associated with increased urine output and GFR in patients with septic shock [25]. Catecholamine and vasopressin were used more frequently in the altered renal function group in this study. In critically ill patients in particular, the degree and severity of systemic inflammation and changes in pathology over time are closely related to renal function. With optimal treatment, sepsis-associated AKI is expected to improve not only the pathogenesis of sepsis, however renal function as well [20,24,27]. Therefore, early TDM should be considered during withdrawal from shock and upon the initiation or discontinuation of circulatory agonists, as changes in VCM blood levels due to changes in renal function can be expected. More attention should be paid to the possibility that steady-state AUC values may deviate from simulations, especially in patients with a 20% or more increase (or decrease) in CCr since the initial TDM.

We recommend AUC estimation using trough and peak concentrations, especially in critically ill patients and in cases where PK variability is high or changes in renal function are expected. To evaluate the accuracy of the AUC, Oda et al. [13] compared the reference AUC obtained by the trapezoidal method using an eight-point blood draw with the AUC obtained by a Bayesian estimation. The AUC estimation using trough and peak concentrations produced the least bias in patients with VCM, administered at 12 h intervals. In contrast, AUC estimation using only the trough produced moderate and evident biases in patients with VCM administered at 12 and 24 h intervals, respectively. Considering the decrease in accuracy, AUC estimation using only the trough concentration should be avoided in the treatment of critically ill patients whose pharmacokinetics show a large variability [28] and for patients with kidney dysfunction who are likely to be prescribed once-daily dosing.

In addition, Hashimoto et al. [10] reported on patients who were candidates for AUC-guided dosing to reduce the risk of AKI. Their study identified the following risk factors for AKI in patients treated with VCM of reduced renal function: tazobactam/piperacillin use, diuretics use, and ICU admission. In this study, which included only patients admitted to the ICU, the SOFA score was significantly higher in the two-point group, and the percentage of diuretic use was additionally higher. The SOFA score is a widely used score for assessing systemic organ failure in critically ill patients and is associated with the development of AKI and mortality [18,29,30]. This indicates that in the two-point group is a more severely ill patient. In fact, a comparison of the incidence of AKI between the 1-point and 2-point blood collection groups showed no significant difference (25% vs. 32%, *p* = 0.442). In other words, AUC estimation by trough and peak values can achieve the target AUC without worsening the development of AKI, even in critically ill patients at a high risk of developing AKI. This supports the previous report, that AUC estimation by two-point blood collection is more accurate than AUC estimation by trough values alone, and the accuracy of this estimation is maintained even in critically ill patients.

The population model of this study may deviate from the population model of the simulation software because the subjects of this study were patients admitted to the ICU. To resolve this issue, we compared the population in this study with the population model created by Yasuhara et al. [31]. Regarding body weight, a difference of approximately 10 kg was observed compared to the model of Yasuhara et al. This difference may be attributed to gender differences. In terms of mean creatinine, the population model from Yasuhara et al. showed higher values; however, the results for CCr were similar between the two populations. However, in this study, we found that changes in renal function over time affected blood levels of VCM. In fact, 19 of the 34 cases in which the CCr between the initial and follow-up TDM changed by a 20% or more increase (or decrease) did not achieve the target AUC. When simulation software is used for critically ill patients, subsequent changes in pathology and renal function may cause the estimated AUC values simulated in the initial dosing design and initial TDM to deviate from the actual measured AUC values.

This present study had several limitations. First, this was a retrospective observational study with a small sample enrolled at a single institution; therefore, selection bias could not be excluded. Second, we believe that the simulation software used in this study, VCM TDM software PAT ver2.1, has limited data for simulating patients admitted to the ICU, and further data are needed to improve the simulations. Third, the new measure in this study, a 20% or more increase (or decrease) in CCr, is not necessarily applicable to all ICU patients. However, it is believed that CCr is easy to calculate and that observing its changes and inferring deviations in blood levels are clinically valuable indicators. We consider that the risk factors identified in this study need to be examined in relation to mortality and AKI, as well as their efficacy against specific bacterial strains.

## 5. Conclusions

AUC evaluation using trough and peak concentrations in patients admitted to the ICU is expected to increase the achievement rate of the target AUC 400–600 μg·h/mL at the TDM follow-up. In addition, we revealed that in patients admitted to the ICU, changes in renal function with a 20% or more increase (or decrease) are a predictor of AUC deviation. In particular, we recommend an early follow-up TDM and AUC evaluation by two-point concentrations (trough/peak) in patients with a fluctuating renal function because a lower AUC achievement rate is expected.

## 6. Clinical Recommendations

The use of two points for the AUC estimation of vancomycin, trough and peak concentrations, increases the accuracy of the estimation;For patients admitted to the ICU in particular, where inter-individual variability is high, AUC estimation using a two-point blood collection is necessary;Changes in renal function are predictors of AUC deviation, and caution should be exercised for changes in CCr with a 20% or more increase (or decrease).

## Figures and Tables

**Figure 1 antibiotics-12-01113-f001:**
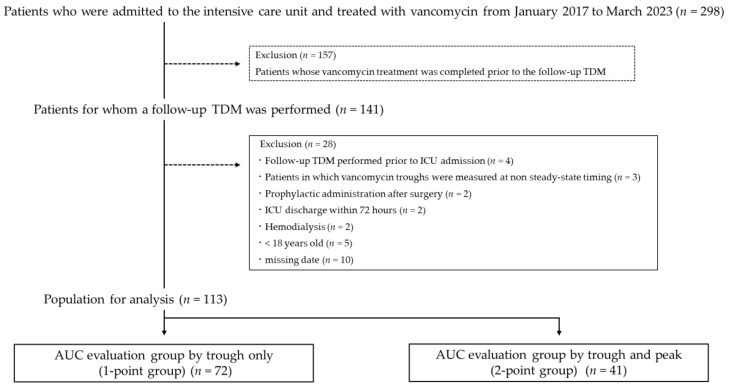
Flowchart of the inclusion of study subjects. ICU, intensive care unit; TDM, therapeutic drug monitoring; and AUC, area under the concentration-time curve.

**Figure 2 antibiotics-12-01113-f002:**
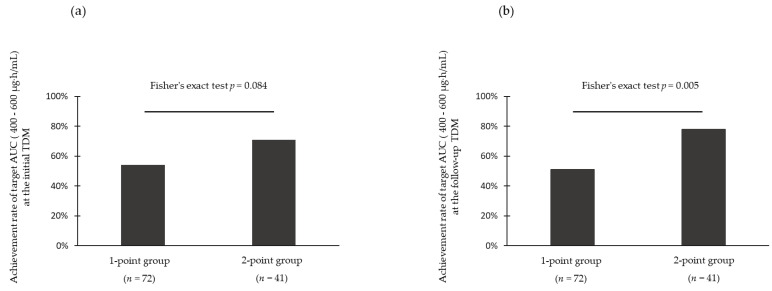
Comparison of the target AUC values at the follow-up TDM by 1-point and 2-point groups. (**a**) Comparison of target AUC at the initial TDM by 1-point (trough concentration only) and 2-point (trough/peak concentration) groups. (**b**) Comparison of target AUC at the follow-up TDM by 1-point (trough concentration only) and 2-point (trough/peak concentration) groups. The target AUC achievement rate was compared by the actual measured AUC values for the two groups, in which the AUC was estimated by 1-point blood collection (trough concentration only) or 2-point blood collection (trough/peak concentration) at the initial TDM. Further, *p* < 0.05 was considered statistically significant. Abbreviations: AUC, area under the concentration–time curve; TDM, therapeutic drug monitoring.

**Figure 3 antibiotics-12-01113-f003:**
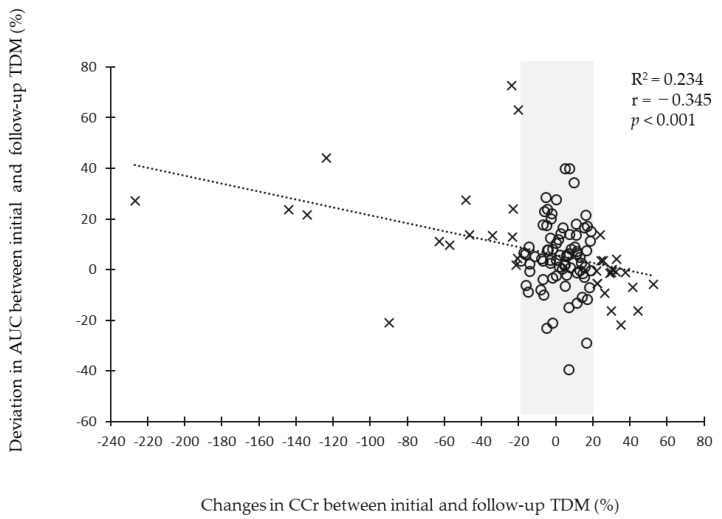
The relationship between percent change in renal function and AUC deviation. The correlation between the rate of changes in renal function, from the initial TDM and follow-up TDM and the rate of deviation of AUC at the initial TDM and AUC at the follow-up TDM. Further, *p* < 0.05 was considered statistically significant. Abbreviations: AUC, area under the concentration–time curve; CCr, creatinine clearance; CRRT, continuous renal replacement therapy; and TDM, therapeutic drug monitoring.

**Table 1 antibiotics-12-01113-t001:** Baseline characteristics of patients.

	All (*n* = 113)	1-Point Group (*n* = 72)	2-Point Group (*n* = 41)	*p* Value
Age, years	64.8 ± 13.0	64.5 ± 14.6	65.1 ± 9.7	0.724
Female, *n* (%)	33 (29.2%)	19 (26.3%)	14 (34.1%)	0.383
Height, cm	162.7 ± 9.4	162.5 ± 9.8	163.0 ± 8.9	0.802
Body weight, kg	61.9 ± 13.2	62.3 ± 14.4	61.2 ± 11.0	0.749
BMI, kg/m^2^	23.3 ± 4.1	23.4 ± 4.3	22.9 ± 3.8	0.772
>25, kg/m^2^	30 (26.5%)	18 (25.0%)	12 (29.2%)	0.621
<18.5, kg/m^2^	10 (8.9%)	5 (6.9%)	5 (12.2%)	0.345
SOFA score, point	7 (4, 11)	7 (3, 9)	9 (6, 11)	0.037
APACHE II score, point	21 (15, 28)	20 (12, 27)	22 (17, 28)	0.073
**Focus of infection, *n* (%)**				0.593
Bacteremia	37 (32.7%)	26 (36.1%)	11 (26.8%)	
Skin and soft tissue	19 (16.8%)	12 (16.6%)	7 (17.1%)	
Abdomen	19 (16.8%)	9 (12.5%)	10 (24.4%)	
Respiratory tract	13 (11.5%)	9 (12.5%)	4 (9.8%)	
CRBSI	8 (7.1%)	4 (5.6%)	4 (9.8%)	
Febrile neutropenia	5 (4.4%)	2 (2.8)	3 (7.3)	
Device-related	3 (2.7%)	2 (2.8%)	1 (2.4)	
Meningitis	3 (2.7%)	3 (4.2%)	0 (0%)	
Bone	2 (1.8%)	2 (2.8%)	0 (0%)	
Urinary tract	1 (0.9%)	1 (1.4%)	0 (0%)	
Fever of unknown organ	3 (2.7%)	2 (2.8%)	1 (2.4)	
**Complications, *n* (%)**				
Hypertension	59 (52.2%)	33 (45.8%)	26 (63.4%)	0.072
Diabetes mellitus	33 (29.2%)	22 (30.6%)	11 (26.8%)	0.675
Dyslipidemia	24 (21.1%)	16 (22.2%)	8 (19.5%)	0.735
**Laboratory data**				
Albumin, g/dL	2.2 (1.9, 2.6)	2.1 (1.9, 2.6)	2.2 (2.0, 2.5)	0.788
Creatinine, mg/dL	0.8 (0.6, 1.2)	0.7 (0.5, 1.1)	0.8 (0.7, 1.2)	0.068
CCr at the vancomycin initiation, mL/min	73.3 (44.8, 104.9)	75.7 (53.7, 106.8)	62.7 (41.9, 100.7)	0.159
CCr at the initial TDM, mL/min	79.5 (51.1, 110.9)	81.0 (56.6, 113.3)	64.9 (40.5, 107.9)	0.201
CCr at the follow-up TDM, mL/min	84.0 (45.3, 108.9)	89.1 (51.8, 118.9)	73.2 (42.8, 100.7)	0.197
% CCr, mL/min	4.48 (−7.03, 15.46)	4.62 (−6.42, 15.83)	3.70 (−7.50, 15.46)	0.905
20% or more increase, *n* (%)	18 (16%)	11 (15%)	7 (17%)	0.802
20% or more decrease, *n* (%)	16 (14%)	9 (13%)	7 (17%)	0.503
eGFRcre, mL/min/1.73 m^2^	71.2 (43.9, 98.3)	74.5 (47.5, 101.3)	55.7 (40.5, 79.5)	0.065
eGFRcre < 60 mL/min/1.73 m^2^, *n* (%)	44 (38.9%)	28 (38.9%)	16 (39.0%)	0.989
WBC, ×10^3^/μL^3^	10.1 (6.4, 14.7)	10.0 (6.4, 14.9)	10.8 (6.3, 14.2)	0.788
CRP, mg/dL	10.3 (5.1, 16.1)	10.9 (5.3, 16.1)	9.4 (4.3, 16.8)	0.511
BUN, mg/dL	26 (17, 38)	21 (16, 34)	30 (24, 55)	0.005
BUN/Creatinine	30 (21, 43)	27 (21, 40)	35 (26, 52)	0.107
**Concomitant, *n* (%)**				
Diuretics	60 (53.1%)	33 (45.8%)	27 (65.9%)	0.040
Iodine-based contrast media	57 (50.4%)	37 (51.4%)	20 (48.8%)	0.789
Catecholamine	56 (49.6%)	29 (40.3%)	27 (65.9%)	0.009
NSAIDs	40 (35.4%)	33 (45.8%)	7 (17.1%)	0.002
Tazobactam/Piperacillin	21 (18.6%)	15 (20.8%)	6 (14.6%)	0.415
Trimethoprim/Sulfamethoxazole	18 (15.9%)	15 (20.8%)	3 (7.3%)	0.059
**Antibiotics**				
Carbapenem	37 (32.7%)	28 (38.9%)	9 (22.0%)	0.065
Cephalosporin	17 (15.0%)	12 (16.7)	5 (12.1%)	0.523
Penicillin (excluding Tazobactam/Piperacillin)	2 (1.8%)	2 (2.8%)	0 (0%)	0.282
Others	1 (0.9%)	1 (1.3%)	0 (0%)	0.449
**Vancomycin therapy**				
Loading dose, *n* (%)	30 (27%)	11 (15.3%)	19 (46.3%)	<0.001
Maintenance dose to initial TDM, mg/kg/day	13.9 (10.4, 16.5)	14.2 (11.1, 16.7)	12.6 (9.0, 16.2)	0.215
Maintenance dose after the initial TDM, mg/kg/day	12.6 (9.5, 16.2)	13.1 (9.9, 16.3)	12.0 (8.4, 16.2)	0.339
Days to initial TDM, day	2 (1, 2)	2 (2, 3)	1 (1, 2)	<0.001
Days from initial TDM to follow-up TDM, day	4 (4, 5)	4 (4, 5)	4 (3, 4)	<0.001
**Dosing interval for vancomycin to initial TDM, *n* (%)**				0.296
8 h	4 (3.5%)	2 (2.8%)	2 (4.9%)	
12 h	84 (74.3%)	57 (79.2%)	27 (65.9%)	
24 h	25 (22.1%)	13 (18.1%)	12 (29.3%)	
**Dosing interval for vancomycin after the initial TDM, *n* (%)**				
8 h	8 (7.1%)	8 (11.1%)	0 (0%)	0.072
12 h	80 (70.8%)	50 (69.4%)	30 (7.1%)	
24 h	25 (22.1%)	14 (19.4%)	11 (26.9%)	
**AUC parameter (μg·h/mL)**				
AUC at the initial TDM	486 (395, 544)	497.1 (394.4, 578.9)	452.0 (397.6, 510.9)	0.031
AUC at the follow-up TDM	492 (426, 574)	524.6 (428.1, 615.7)	476.6 (420.2, 548.0)	0.115

Data are presented as mean (standard deviation [SD]), or median (interquartile range [IQR] 25th–75th percentile), or numbers (with percentages). *p* < 0.05 was considered statistically significant. Abbreviations: BMI, body mass index; SOFA, sequential organ failure assessment; APACHE II, acute physiology and chronic health evaluation; CRBSI, catheter-related blood stream infection; CCr, creatinine clearance; WBC, white blood cell; CRP, C-reactive protein; AUC, area under the curve; TDM, therapeutic drug monitoring; and BUN, blood urea nitrogen.

**Table 2 antibiotics-12-01113-t002:** Logistic regression analysis of factors related to achievement of the target AUC.

	Univariate Model			Multivariate Model		
	OR	(95% CI)	*p* Value	OR	(95% CI)	*p* Value
Age; per 1-year increase	1.00	0.97–1.03	0.848	1.00	0.97–1.04	0.901
Sex; female	1.23	0.54–2.80	0.626	0.99	0.35–2.83	0.984
BMI; ≥25 kg/m^2^	0.54	0.23–1.25	0.150			
APACHE II score; per 1-point increase	1.00	0.96–1.05	0.909			
SOFA score; per 1-point increase	1.02	0.95–1.11	0.519			
Sepsis; yes	0.71	0.33–1.53	0.385			
Burns; yes	0.23	0.04–1.26	0.090			
Loading dose; yes	2.11	0.84–5.28	0.112			
2-point group; yes	3.36	1.41–8.04	0.006	2.89	1.06–7.84	0.038
20% or more increase (or decrease) in CCr; yes	0.37	0.16–0.84	0.017	0.25	0.09–0.67	0.006
Days from initial TDM to follow-up TDM; >4 days	0.33	0.15–0.75	0.008	0.44	0.18–1.09	0.075
CRRT; yes	0.55	0.23–1.29	0.169			
Catecholamine; yes	1.13	0.53–2.40	0.756			
Diuretics; yes	2.25	1.04–4.86	0.040	2.25	0.95–5.37	0.067
Tazobactam/Piperacillin; yes	1.78	0.63–4.95	0.284			

*p* < 0.05 was considered statistically significant. Abbreviations: OR, odds ratio; CI, confidence interval; BMI, body mass index; APACHE II, acute physiology and chronic health evaluation; CCr, creatinine clearance; and CRRT, continuous renal replacement therapy.

**Table 3 antibiotics-12-01113-t003:** Characteristics of patients with or without changes in CCr.

	Without-Change Group (*n* = 79)		With-Change Group (*n* = 34)		*p* Value
Age, years	64.6 ± 12.3		65.2 ± 14.6		0.932
Female, *n* (%)	17 (22%)		16 (47%)		0.006
BMI, kg/m^2^	23.0 ± 3.9		23.9 ± 4.7		0.507
SOFA score, point	7 (3, 10)		8 (6, 11)		0.169
APACHE II score, point	20 (12, 25)		22 (17, 28)		0.144
Sepsis, *n* (%)	37 (47%)		24 (71%)		0.020
Septic shock, *n* (%)	28 (35%)		19 (56%)		0.043
CRRT, *n* (%)	15 (19%)		13 (38%)		0.035
**Complications, *n* (%)**					
Hypertension	42 (53%)		17 (50%)		0.757
Diabetes mellitus	23 (29%)		10 (29%)		0.975
Dyslipidemia	16 (20%)		8 (24%)		0.677
**Laboratory data**					
Creatinine, mg/dL	0.7 (0.6, 1.0)		1.0 (0.6, 1.5)		0.018
CCr at the vancomycin initiation, mL/min	76.4 (59.9, 108.3)		49.6 (33.0, 87.5)		0.015
CCr at the initial TDM, mL/min	82.3 (58.3, 122.5)		58.7 (43.8, 92.4)		0.024
CCr at the follow-up TDM, mL/min	87.8 (60.9, 122.8)		61.2 (37.2, 98.6)		0.032
% CCr, mL/min	3.70 (−4.76, 11.25)		22.08 (−46.98, 30.24)		0.621
eGFRcre, mL/min/1.73 m^2^	76.8 (52.2, 101.4)		53.3 (30.4, 72.9)		0.004
eGFRcre<60 mL/min/1.73 m^2^, *n* (%)	27 (34%)		17 (50%)		0.114
WBC, ×10^3^/μL	10.4 (6.8, 13.6)		9.1 (5.8, 15.0)		0.606
CRP, mg/dL	10.1 (5.0, 16.0)		12.0 (5.9, 18.1)		0.719
BUN, mg/dL	22 (16, 34)		34 (22, 54)		0.003
BUN/Creatinine, mg/dL	29 (20, 42)		31 (23, 49)		0.472
**Concomitant, *n* (%)**					
Diuretics	39 (49%)		21 (62%)		0.226
Catecholamine	34 (43%)		22 (65%)		0.035
Vasopressin	4 (5%)		7 (21%)		0.011
Tazobactam/Piperacillin	17 (22%)		4 (12%)		0.222
**Vancomycin therapy**					
Loading dose, *n* (%)	21 (27%)		9 (26%)		0.990
Maintenance dose to initial TDM, mg/kg/day	13.9 (10.2, 16.7)		14.2 (10.6, 15.8)		0.684
Maintenance dose after initial TDM, mg/kg/day	13.2 (9.1, 16.3)		11.9 (10.3, 16.1)		0.719
Days to initial TDM, day	2 (1, 2)		2 (1, 3)		0.906
Days from initial TDM to follow-up TDM, day	4 (4, 5)		4 (4, 5)		0.759
**AUC parameter (μg·h/mL)**					
AUC at the initial TDM	445.8 (378.1, 523.2)		408.6 (330.3, 585.9)		0.637
AUC at the follow-up TDM	495.8 (442.1, 561.9)		464.5 (391.2, 616.3)		0.712

Data are presented as mean (standard deviation [SD]), or median (interquartile range [IQR] 25th–75th percentile), or numbers (with percentages). Further, *p* < 0.05 was considered statistically significant. Abbreviations: BMI, body mass index; SOFA, sequential organ failure assessment; APACHE II, acute physiology and chronic health evaluation; CCr, creatinine clearance; WBC, white blood cell; CRP, C-reactive protein; CRRT, continuous renal replacement therapy; AUC, area under the curve; TDM, therapeutic drug monitoring; and BUN, blood urea nitrogen.

## Data Availability

The datasets generated and/or analyzed during this current study are not publicly available because a research agreement from all authors is required for data sharing, however are available from the corresponding author on reasonable request.

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
