# Peer review of "Simulation of Vancomycin Exposure Using Trough and Peak Levels Achieves the Target Area under the Steady-State Concentration–Time Curve in ICU Patients"

_antibiotics, 2023, doi:10.3390/antibiotics12071113_

Round 1
Reviewer 1 Report
The article discusses the factors influencing achievement of the target area under the concentration time curve (AUC) of vancomycin (VCM) at steady state in critically ill patients. The study found that using both trough and peak VCM concentrations in TDM evaluation was significantly associated with the success rate of achieving the target AUC at steady state for VCM. Changes in renal function observed in critically ill patients were identified as a novel risk factor for reduced attainment. The authors recommend AUC evaluation of VCM with 2-point blood collection, close monitoring of renal function, and dose adjustment based on reanalyzing the blood concentrations of VCM.
The manuscript is well-written and well-organized. I recommend accepting it pending a thorough review for any necessary text editing.
Minor editing of English language required
Author Response
Thank you for reviewing our manuscript and for the constructive comments provided, which have helped us improve the manuscript. We have revised the manuscript according to reviewers' comments. The revised manuscript has been edited by a native English speaker.
Reviewer 2 Report
The manuscript written in the title of “Type of the PaperUsing trough and peak vancomycin concentrations in critically ill patients achieves the target area under the concentration time curve at the steady state”, the authors should consider the followings:
1. The authors should design the study with the use of a more direct primary outcome.
2. In statistical analysis, the authors should justify why they use Youden index, instead of other alternatives.
3. Why the authors only use VCM TDM software PAT ver2.1, than the other available softwares? Would the authors draw a different study outcomes from a different software?
4. Please justify why the study narrowed the time of records from Jan 2017 to March 2023, instead of a longer period of records.
5. The authors should state the current practice of TDM in the study (by the hospital) and introduce (if available) their suggested recommendation for that in a clear manner. (i.e. using a workflow or table to present the treatment plan)
6. The authors may investigate the types of diuretics and Catecholamine, taken by the patients.
7. Please clarify the novelty of this study in the Abstract and conclusion part of the article.
8. The authors should review their use of English in this article.
9. The authors may revise their title of the article to clearly represent their captioned study.
10. In methodology of the section of VCM dosing and pharmacodynamics data, the authors should show the equations of eGFRcre.
11. As of Table S1, the state specificity were rather low, the authors should elaborate more on that.
12. In methodology (outcomes) sections, the authors should list all the items in secondary outcomes.
13. The authors should provide the LLOQ and measurement uncertainty of the analytes measured in the study.
14. In Figure 2, the authors should show the error bars in the Figure 2a and 2b.
15. Please show R square value of Figure 3, and explain the values.
16. The authors should elaborate clearly on the several limitations of the present study.
English rather difficult to understand
Author Response
Thank you for reviewing our manuscript and for the constructive comments provided, which have helped us improve the manuscript. We have revised the manuscript according to reviewers’ comments. The changes using “Track Changes” have been made in the text. Our point-by-point responses are given below.
- The authors should design the study with the use of a more direct primary outcome.
Author’s response
The primary outcome of this study was to identify novel risk factors associated with achieving the target AUC for VCM, focusing only on critically ill patients whose blood levels were difficult to predict. We believe that such identification of novel risk factors will help prevent acute kidney injury caused by VCM. We also believe that controlling VCM blood levels at the targeted AUC will ensure efficacy for patients admitted to the ICU. We believe that the risk factors identified in this study need to be examined in the future to determine their relevance to efficacy and safety. Therefore, we have added information regarding the limitations of this study (p. 12, “Limitations”).
- In statistical analysis, the authors should justify why they use Youden index, instead of other alternatives.
Author’s response
The Youden index is used as an indicator of cutoff values. The other available method entails taking the distance from the upper left of the figure of the receiver operating characteristic curve with sensitivity on the vertical axis and 1 - specificity on the horizontal axis. As you pointed out, the required sensitivity and specificity may vary depending on the purpose of the research; however, the Youden index, which is commonly employed, was used to determine the cutoff value in this study.
- Why the authors only use VCM TDM software PAT ver2.1, than the other available softwares?Would the authors draw a different study outcomes from a different software?
Author’s response
As this study was conducted in a sample of Japanese patients, TDM was performed using the VCM TDM software PAT ver2.1, which was created with Japanese patients in mind. Comparisons with other TDM software programs are currently under consideration.
- Please justify why the study narrowed the time of records from Jan 2017 to March 2023, instead of a longer period of records.
Author’s response
The study period was defined as from the time that pharmacists intervened in the ICU wards and started VCM dosing designed according to the Japanese guidelines. This was because most VCM dosing designs prior to that time had a fixed dose of 1000 mg/q12h, which may have affected the results of the analysis and were therefore excluded from the study period.
- The authors should state the current practice of TDM in the study (by the hospital) and introduce (if available) their suggested recommendation for that in a clear manner. (i.e. using a workflow or table to present the treatment plan)
Author’s response
We have added the status of vancomycin TDM at our hospital in the methods section (p.2-3, Methods “Study design and participants”).
We also believe that early follow-up TDM intervention for patients with CCr changes of more than ±20% or with specific factors related to changes in renal function, such as those obtained in this study, may lead to the control of the target AUC value and prevention of AKI. Specifically, follow-up TDM is typically re-evaluated approximately 1 week after the initial TDM; however, for patients with presumed changes in renal function, we propose early TDM on days 2–3. Specific recommendations that may be suggested based on the results of this study have been added to the "Clinical Recommendations" section (p.13, “Clinical recommendations”).
- The authors may investigate the types of diuretics and Catecholamine, taken by the patients.
Author’s response
As per your suggestion, we further investigated the types of diuretics and catecholamines and have added the detailed types to the results section of our revised manuscript (p.4, Results “Study participant comparison”).
- Please clarify the novelty of this study in the Abstract and conclusion part of the article.
Author’s response
Thank you for this suggestion. We have revised the abstract and conclusion sections of the article, as you indicated (p.1, “Abstract”; p.12, “Conclusion”).
- The authors should review their use of English in this article.
Author’s response
This article has been checked by a native English speaker and re-edited before resubmission. A certificate of English editing is attached.
- The authors may revise their title of the article to clearly represent their captioned study.
Author’s response
In accordance with your suggestion, we have changed the title to "Simulation of vancomycin exposure using trough and peak levels achieves the target area under the steady-state concentration-time curve in ICU patients".
- In methodology of the section of VCM dosing and pharmacodynamics data, the authors should show the equations of eGFRcre.
Author’s response
Thank you for your suggestion, we have added the equation for eGFRcre in the VCM dosing and pharmacodynamics data section (p.3, Methods “VCM dosing and pharmacodynamics data”).
- As of Table S1, the state specificity were rather low, the authors should elaborate more on that.
Author’s response
We believe that changes in CCr of 20% or more increase (or decrease), identified as a novel risk factor in this study, are easy to use in clinical practice. However, as you have pointed out, it is not necessarily applicable to all ICU patients because of its low specificity. Therefore, we should also be aware of cases with specific factors, as indicated in Table 3, even if the renal function does not deviate. We have added explanation to the discussion and study limitations sections, as these were not fully explained (p.11, “Discussion”; p.12, “Limitations”).
- In methodology (outcomes) sections, the authors should list all the items in secondary outcomes.
Author’s response
Owing to the lack of information listed under Secondary Outcomes, the items considered in this study have been added to the Methodology (Outcomes) section (p.3, Methods “Outcomes”).
- The authors should provide the LLOQ and measurement uncertainty of the analytes measured in the study.
Author’s response
Thank you for your suggestion. We have added to the methods section the LLOQ and measurement uncertainty of the analyte measured in this study (p.2, Methods “VCM dosing and pharmacodynamics data”).
- In Figure 2, the authors should show the error bars in the Figure 2a and 2b.
Author’s response
In Figure 2, the error bars are not shown because the target AUC for vancomycin is shown as percent achieved (percentage).
- Please show R square value of Figure 3, and explain the values.
Author’s response
The R-squared value has been added to Figure 3. We have also added an explanation of the correlation to the results in the revised manuscript (p.7, Results “Relationship between changes in renal function and area under the curve deviation”).
- The authors should elaborate clearly on the several limitations of the present study.
Author’s response
Based on your comments regarding the results of this study, we added several limitations to our revised discussion section.
Reviewer 3 Report
Greetings to the authors! Congratulations on a well done work and the preparation of the manuscript. I have several suggestions.
1. I propose to correct the title of the manuscript, based on the principle of "first things first", in a more laconic way, for example, to: "Vancomycin concentration in blood of ICU patients during 1 and 2-point blood collection by the area under the concentration-time curve estimation".
2. In the abstract, it is worth mentioning that 2-point checking means 12 and 24 hours after hospitalization, if I understood correctly.
3. Why is the p-error (p) not calculated for all indicators in Table 1? For example, for the focus of infection.
4. In the notes to Table 1, explain all abbreviations, for example, the abbreviation "BUD". Yes, the transcript is at the end of the manuscript, then under the table, indicate that the transcript is at the end.
5. Is Figure 2 data mean those patients who, at the beginning of treatment, have better indicators, then better achieve the target levels of vancomycin concentration in 24 hours? Did I understand correctly? Please write a more accessible explanation.
6. Were patients who had a co-infection with Gram (+) and Gram (-) pathogens and accordingly received several antibiotics considered? If not, please note that.
7. When you say "CCr of more than ±20%," do you mean possible variation +20% and -20% as well? In Table 3 CCr lever changing describes in percentile, not in %, so it is unclear where you get 20%. And "CCr of more than ±20%", mean -20%, not `+`, as I understood, so sign “±” less relevant. Please, clarify it.
8. I think it's worth highlighting "clinical recommendations" in a separate section or a separate table, and information about the need for 2-point checking of the vancomycin level should be indicated in a separate section because this information is lost in the general discussion.
9. In the discussion, it is worth briefly mentioning the pharmacokinetics of Vancomycin and why measuring its concentration at the 12th hour is crucial.
Author Response
Thank you for reviewing our manuscript and for the constructive comments provided, which have helped us improve the manuscript. We have revised the manuscript according to reviewers’ comments. The changes using “Track Changes” have been made in the text. Our point-by-point responses are given below.
- I propose to correct the title of the manuscript, based on the principle of "first things first", in a more laconic way, for example, to: "Vancomycin concentration in blood of ICU patients during 1 and 2-point blood collection by the area under the concentration-time curve estimation".
Author’s response
Thank you for this suggestion. As you suggested, we have changed the title to "Simulations of vancomycin exposure using trough and peak levels achieves the target area under the steady-state concentration-time curve in ICU patients".
- In the abstract, it is worth mentioning that 2-point checking means 12 and 24 hours after hospitalization, if I understood correctly.
Author’s response
The 2-point blood draw in this study refers to the trough value immediately before VCM administration and the peak value 1–2 h after administration.
- Why is the p-error (p) not calculated for all indicators in Table 1? For example, for the focus of infection.
Author’s response
In Table 1, the analysis of "focus of infection" was performed using a 2 × 11 Chi-square test, assuming no overlapping infections occurred during the same period, yielding a p-value of 0.593 (Table 1). A 2 × 3 Chi-square test was also performed for “Dosing Interval for vancomycin to initial TDM” and “Dosing Interval for vancomycin after the initial TDM” with p-values of 0.296 and 0.072, respectively.
- In the notes to Table 1, explain all abbreviations, for example, the abbreviation "BUD". Yes, the transcript is at the end of the manuscript, then under the table, indicate that the transcript is at the end.
Author’s response
Thank you for pointing this out. We have added the abbreviations to Table 1.
- Is Figure 2 data mean those patients who, at the beginning of treatment, have better indicators, then better achieve the target levels of vancomycin concentration in 24 hours? Did I understand correctly? Please write a more accessible explanation.
Author’s response
Figure 2a shows the actual AUC at the time of blood collection on days 1–4 after the first VCM dose was administered. The AUC was calculated for each of the two groups: 1-point blood collection (trough concentration only) and 2-point blood collection (trough/peak concentration), and the percentages of target AUC achieved were compared. Meanwhile, Figure 2b shows the measured AUC at steady state 2–8 days after the initial TDM. The target AUC achievement rate was compared with the actual measured values for the two groups, in which the AUC was estimated by 1-point blood collection (trough concentration only) or 2-point blood collection (trough/peak concentration) at the initial TDM. As these explanations were lacking, supplemental explanations have been added to Figure 2.
- Were patients who had a co-infection with Gram (+) and Gram (-) pathogens and accordingly received several antibiotics considered? If not, please note that.
Author’s response
As you indicated, mixed infections with Gram-positive and Gram-negative pathogens were also included. Therefore, we also included cases where antibiotics were administered in combination with VCM. This has been added to the manuscript, and concomitant antimicrobials are further described in Table 1 (p.4, Results “Study participant comparison”).
- When you say "CCr of more than ±20%," do you mean possible variation +20% and -20% as well? In Table 3 CCr lever changing describes in percentile, not in %, so it is unclear where you get 20%. And "CCr of more than ±20%", mean -20%, not `+`, as I understood, so sign “±” less relevant. Please, clarify it.
Author’s response
Thank you for pointing this out. Changes in CCr +20% were defined as a 20% or more improvement in CCr from the time of the initial TDM to follow-up TDM. In contrast, changes in CCr of -20% or more refer to a CCr worsening by 20%. The description has been changed to "20% or more increase (or decrease) in CCr" because "changes in CCr of more than ±20%" could be misleading. Since these explanations were not sufficient, we supplemented them in the result section (p.7, Results “Relationship between changes in renal function and area under the curve deviation”). In addition, we have shown the values of CCr and %CCr at each time point in Tables 1 and 3.
- I think it's worth highlighting "clinical recommendations" in a separate section or a separate table, and information about the need for 2-point checking of the vancomycin level should be indicated in a separate section because this information is lost in the general discussion.
Author’s response
Thank you for your suggestion. We have included a separate section on the need for 2-point blood draws for AUC estimation of vancomycin in ICU patients in a revised "Clinical Recommendations" section (p.13, “Clinical recommendations”).
- In the discussion, it is worth briefly mentioning the pharmacokinetics of Vancomycin and why measuring its concentration at the 12th hour is crucial.
Author’s response
Unless I am mistaken, based on my interpretation, this study did not address the measurement of concentrations after the 12th hour. I have added a note to the discussion section about the pharmacokinetics of vancomycin (p.11 “Discussion”).